# Detection of Coal and Gangue Based on Improved YOLOv8

**DOI:** 10.3390/s24041246

**Published:** 2024-02-15

**Authors:** Qingliang Zeng, Guangyu Zhou, Lirong Wan, Liang Wang, Guantao Xuan, Yuanyuan Shao

**Affiliations:** 1College of Mechanical and Electronic Engineering, Shandong University of Science and Technology, Qingdao 266590, China; qlzeng@sdust.edu.cn (Q.Z.); 13619970136@126.com (G.Z.); wanlr666@163.com (L.W.); 2College of Mechanical and Electrical Engineering, Shandong Agricultural University, Taian 271018, China; xuangt@sina.com

**Keywords:** coal, gangue, identification, YOLOv8, lightweight

## Abstract

To address the lightweight and real-time issues of coal sorting detection, an intelligent detection method for coal and gangue, Our-v8, was proposed based on improved YOLOv8. Images of coal and gangue with different densities under two diverse lighting environments were collected. Then the Laplacian image enhancement algorithm was proposed to improve the training data quality, sharpening contours and boosting feature extraction; the CBAM attention mechanism was introduced to prioritize crucial features, enhancing more accurate feature extraction ability; and the EIOU loss function was added to refine box regression, further improving detection accuracy. The experimental results showed that Our-v8 for detecting coal and gangue in a halogen lamp lighting environment achieved excellent performance with a mean average precision (mAP) of 99.5%, was lightweight with FLOPs of 29.7, Param of 12.8, and a size of only 22.1 MB. Additionally, Our-v8 can provide accurate location information for coal and gangue, making it ideal for real-time coal sorting applications.

## 1. Introduction

Coal is one of the most important energy sources in the world. However, its mining and processing generate gangue, a waste material that poses significant challenges to the environment. The mixed coal–gangue not only reduces combustion efficiency but also generates harmful gases, severely polluting the environment. Therefore, accurate gangue detection and separation are of great importance.

Traditional methods include manual picking, wet selection, and dry selection. Manual picking relies on the worker’s experience and involves a poor working environment, low efficiency, and low accuracy. The wet separation method can pick out gangue and coal with different calorific values by the density difference, but with the disadvantage of water waste and environmental pollution. Dry separation methods like spectroscopy and X-ray diffraction, while promising, raise concerns about radiation exposure. Machine vision offers a promising alternative. And with advancements in artificial intelligence, deep learning algorithms are increasingly employed for coal and gangue identification. Zhang et al. [1] compared the YOLOv4, SSD, and Faster R-CNN algorithms to detect coal gangue and drew the conclusion that the YOLOv4 detection algorithm performed better with an mAP value of 97.52%. Junpeng Zhou et al. [2] presented an improved BASA-LS-FSVM classification algorithm to separate gangue from coal, with separation accuracy reaching 98%. Hu feng et al. [3] constructed a principal component analysis network (PCANet) model on multispectral images to identify coal and gangue. Xue et al. [4] studied a lightweight Yolo coal gangue detection algorithm based on the resnet18 backbone feature network, with a speed of 45.5 ms/piece and a model size of only 65.34 MB with an mAP of 96.27%. Hengxuan Luan et al. [5] proposed an image window pixel information interaction method that synergizes the strengths of convolutional neural networks (CNNs) and Swin Transformer and optimized the model, achieving 95% precision. Yan Pengcheng et al. [6] presented an improved YOLOv5 algorithm to detect coal and gangue with a detection average accuracy of 98.34%. Wang Xi et al. [7] proposed a semantic segmentation network of coal and gangue images (SSNet_CG) based on the pyramid scene parsing network (PSPNet), with MPA, mIoU, and F1_scores of 97.3, 95.4, and 0.98, respectively, and a single-image test time of 0.027 s. Ziqi Lv et al. [8] presented a synchronous detection–segmentation method for oversized gangue in a coal preparation plant based on multi-task learning. Wenhao Lai et al. [9] proposed an improved Mask R-CNN algorithm based on multispectral images to segment coal gangue and predict its location and shape. Feng Hu et al. [10] combined multispectral imaging technology with the convolutional neural network (CNN) to classify coal and gangue, in which the hyperparameters of the CNN model were optimized by the Bayesian algorithm. Ziqi Lv et al. [11] used the convolutional neural network (CNN) for online detection of coal and gangue, with a detection accuracy of 91.375. Ziqi Lv et al. [12] proposed a single-shot fine-grained object detector using the attention mechanism and applied it to coal–gangue images in a coal preparation plant with APiou = 0.5. Pu et al. [13] used convolutional neural networks and transfer learning to realize the image recognition of coal gangue. Furthermore, Li et al. [14] established a hierarchical deep learning framework to realize the detection of coal gangue through image recognition.

These studies have achieved good detection accuracy; however, most of them solely focus on object detection, neglecting the crucial location information of coal gangue, which is also vital for subsequent separation operations involving manipulators, air blowing mechanisms, etc. Furthermore, to apply to the actual coal sorting production line, both the detection model size and the detection speed also need to be carefully considered. Additionally, most studies only identify gangue from one type of coal. The images are inherently sensitive to light conditions, adding to the complexity of the detection task. Taking these factors into account, a novel, intelligent coal and gangue detection method needs to be proposed. YOLO (you only look once) has a simple principle and a low requirement for hardware performance. As a major update, YOLOv8 was launched by Ultratics on 10 January 2023, with higher speed and accuracy, which allows it to detect objects in real-time with a high level of precision. In this paper, we present a dataset of images capturing two types of coal (density less than 1.4 g/cm^3^, density greater than 1.4 but less than 1.6 g/cm^3^) and gangue on conveyor belts under two varying light environments. Then we improved the YOLOv8 algorithm to make the detection model lightweight.

## 2. Materials and Methods

### 2.1. Image Acquisition

The coal and gangue blocks used in this experiment were obtained from Shandong Energy Group Co., Ltd., Taian, China, Xinwen mine, and the samples of similar size were divided into three types, a, b, and c, according to their density, as shown in Figure 1. Coal and gangue images were captured, respectively, in the Agricultural Equipment Intelligent Engineering Laboratory of Shandong Agricultural University in Tai’an City, Shandong Province, China, in June 2022. During image collection, 4 to 8 pieces of the samples were randomly placed horizontally on a black conveyor belt, and a Canon EOS 90D camera (Canon, Tokyo, Japan) was used to take images vertical to the conveyor belt. The camera height was set to 60 cm, and the image resolution was set to 3840 × 2160 pixels. As shown in Figure 1, two types of image collection environments were set. The light source was a halogen lamp (HSIA-LS-T-200W, Jiangsu Dualix Spectral Imaging Technology Co., Ltd., Wuxi, Jiangsu, China) for Type 1 images and an indoor fluorescent lamp (36 W) for Type 2 images. Images of the three types of samples were collected, respectively, in each of the two light environments mentioned above. The original images of the Type 1 and Type 2 datasets were 424 and 459 and were expanded to 1274 and 1377 images, respectively, by means of rotation, blurring, noise addition, and brightness change randomly. To adapt to the model input, the sizes of all images were cropped to 640 × 640 pixels. Finally, the dataset was divided into training, validation, and test sets in a ratio of 7:1:2, as shown in Table 1. LableImg 1.8.1, an open-source software, was used to annotate the type and location information of the sample in the image.

### 2.2. Laplacian Image Enhancement Module

In the process of collecting and storing images, images often have noise and blur problems, which will directly affect the performance of the model in recognizing coal types and gangue. Therefore, the Laplacian image enhancement module is used to improve the quality of images before they are input to the CNN model for classification. The Laplacian algorithm is a spatial domain enhancement algorithm that uses second-order differentiation to enhance images. In this study, the detected image was first enhanced by the Laplacian algorithm using a 3 × 3 template and then input to the improved Yolov8 model. The image enhancement process expression is given in Formula (1).
(1)$$g(x,y)=f(x,y)+c[\nabla^{2}f(x,y)]$$
where g(x,y) is the output, f(x,y) is the original image, and c is the coefficient.

The comparison of the image before and after Laplacian enhancement is shown in Figure 2. Comparing the upper image with the lower image, it can be seen that regardless of Type 1 or Type 2 images, after Laplacian enhancement, the existence of the target in the image is more prominent, the contour of coal and gangue is clearer, the target main pixel area is strengthened, and the color anisotropy is more obvious, which helps the convolutional neural network to extract the features of different types of targets from the image.

### 2.3. YOLOv8 Network Model

In order to achieve further lightweight, YOLOv8 replaces the C3 module of YOLOv5 with the C2f module. As shown in Figure 3, YOLOv8 retains the SPPF (spatial pyramid pooling-fast) module used in YOLOv5 and other architectures to achieve multi-scale feature fusion. YOLOv8 also retains the FPN + PAN structure in YOLOv5 and removes the convolution structure in the upsampling stage to reduce the computational complexity. To detect coal and gangue of different sizes, the YOLOv8 network endpoint uses the decoupled head to separate the classification and detection heads to improve detection accuracy and also changes from anchor-based to anchor-free. At the same time, the DFL (distribution focal loss) idea is used to extract more detailed information about the coals and gangues, and the number of channels in the regression head is also changed to 4*reg_max to improve the regression accuracy.

### 2.4. Improvement of YOLOv8

#### 2.4.1. C2f Feature Extraction Module

As shown in Figure 4, the C2f module is designed based on the ideas of the C3 module and the ELANs (efficient layer aggregation networks), the idea of feature extraction from parallel streams in CSPNet (cross-stage partial networks), and the idea of residual structure, allowing YOLOv8 to obtain richer gradient flow information while ensuring lightweight. Here, the C2f module is used to extract coal and gangues features from the input data, which is a lightweight and efficient module that can help to improve the performance of the network.

#### 2.4.2. Convolutional Block Attention Module

The convolutional block attention module (CBAM) is a lightweight and generic mechanism that does not add any additional computational cost to the model (Figure 5). CBAM combines the spatial attention mechanism (SAM) and the channel attention mechanism (CAM). It can select key features for the current task, improving the representation ability of CNN. Spatial attention is used to highlight important spatial locations in the feature maps and channel attention to the specific content [14]. In this study, the CBAM attention mechanism is embedded before the SPPF module, further enhancing the feature extraction ability of the backbone network, which can lead to improved accuracy in coal and gangue detection.

#### 2.4.3. Box Regression Loss Function

The original box regression loss function of YOLOv8 is in the form of CIOU loss + DFL. CIOU loss, which stands for complete intersection over union loss, aims to provide a more accurate and comprehensive measure of the similarity between the predicted bounding box and the ground truth box. To effectively improve the detection accuracy of YOLOv8, the loss function is improved from CIOU to EIOU in this study. EIOU loss [1] is based on the factor of CIOU affecting the aspect ratio and separately calculates the difference between the width and height of the real box and the predicted box. The loss function consists of three parts: IoU loss (L_IOU_), center distance loss (L_dis_), and aspect ratio loss (L_asp_). EIoU retains the advantages of CIOU and minimizes the difference between the width and height of the real box and the predicted box, thereby accelerating the convergence speed and improving the regression accuracy. The calculation formula for EIOU loss is as follows:(2)$$\begin{array}{cc} & L_{EloU}=L_{IOU}+L_{dis}+L_{asp}=1-IoU+\frac{\rho^{2}(b,b^{gt})}{c^{2}}+\frac{\rho^{2}(w,w^{gt})}{c_{w}^{2}}+\frac{\rho^{2}(h,h^{gt})}{c_{h}^{2}} \end{array}$$
where ρ is the Euclidean distance between the predicted box and the real box; b, w, and h are the center point, width, and height of the predicted box, respectively; $b^{gt},w^{gt},h^{gt}$ represent the center point, width, and height of the real box, respectively; $c,c_{w},c_{h}$ represent the diagonal length, width, and height of the smallest bounding rectangle containing the predicted box and the real box, respectively.

The prediction process of CIOU and EIOU loss functions at different iterations is shown in Figure 6. In Figure 6, the blue box is the real box, the black box is the preset anchor box, and the red box and green box are the regression processes of the predicted boxes of CIOU and EIOU, respectively. It can be seen from the figure that the width and height of CIOU cannot be increased or decreased at the same time, while EIOU can.

DFL (distribution focal loss) mainly aims to model the position of the target box as a general distribution, allowing the network to focus more quickly on the values near the target box position, increasing their probability. In this study, DFL is used to optimize the probability of the two positions closest to label y in the form of cross entropy (Figure 7), so that the network can focus more quickly on the distribution of adjacent regions of the target position, which can reduce the complexity of the model and improve the convergence speed. That is to say, the learned distribution theory is located near the real floating-point coordinates, and the weight of the distance between the left and right integer coordinates is obtained through linear interpolation.

### 2.5. Model Experimental Environment and Evaluation Indicators

The computer hardware included an Intel Core i5-8300H CPU @ 2.30 GHz and an NVIDIA GeForce GTX 1080 Ti GPU. The model was trained on a single GPU using Python 3.8 and CUDA 10.2, with an initial learning rate of 0.01. The maximum number of training iterations was set to 50, with a momentum of 0.937. The batch size was set to 48. The performance of the model was evaluated using precision (P), recall (R), average precision (AP), and mean average precision (mAP). The definitions of these metrics are as follows:(3)$$P=\frac{TP}{TP+FP}\times 100\%$$
(4)$$R=\frac{TP}{TP+FN}\times 100\%$$
(5)$$AP=\int_{0}^{1}P(R)dR\times 100\%$$
(6)$$mAP=\frac{\sum_{n=1}^{2}AP_{n}}{2}\times 100\%$$
where TP, FP, and FN represent true positive, false positive, and false negative, respectively, with n representing the nth sample. In addition, to evaluate the computational capability and inference speed of the model, the number of parameters (Params), floating-point operations per second (FLOPs), and frames per second (FPS) were used as evaluation indicators.

## 3. Results and Discussion

### 3.1. Model Training

The improved YOLOv8 model is named Our-v8. Our-v8 is trained on the original and augmented datasets by the Laplacian algorithm, respectively. The training process is shown in Figure 8. As shown in Figure 8a, compared to Our-v8 trained on the original Type 1 and Type 2 datasets (named as Type 1 and Type 2), the mAP curve of Our-v8 trained on the Type 1 and Type 2 datasets processed by the Laplacian algorithm (named as L-Type 1 and L-Type 2) continues to rise with slight fluctuations, and at the end of the model training, Our-v8 trained on the processed data has a higher mAP than Our-v8 trained on original data, and the mAP is all above 0.99. As shown in Figure 8b, the loss curves of L-Type 1 and L-Type 2 decrease rapidly at the beginning of training and show a downward trend with obvious fluctuations. At the end of training, L-Type 1 and L-Type 2 have a lower loss value than Type 1 and Type 2, and the loss value is all below 0.79. The results show that the model trained on the processed dataset has a higher mAP and a lower loss than the model trained on the original dataset. This indicates that the proposed Laplacian augmentation algorithm can effectively improve the accuracy and reduce the loss of the YOLOv8 model.

### 3.2. Comparison of Our-v8 and YOLOv8

In order to compare the performance of the proposed Our-v8 model with the original model, coal and gangue images of the test dataset are used as input to the model, and the performance based on the visualization results of the model feature maps and the detection results are evaluated in this study. A gradient-weighted class activation map (Grad-CAM) is used to visualize the last C2f layer of YOLOv8 and Our-v8 to demonstrate the feature extraction capabilities of the models, respectively. The visualization results are shown in Figure 9, with the left part for dataset Type 1 and the right part for Type 2. The dark blue value is 0, and the red value is 1. The redder the color, the greater the weight of the region in making specific category decisions for the model.

As can be seen from Figure 9, compared to YOLOv8, Our-v8 pays more attention to the coal and gangue areas and is less affected by background factors in Type 1 and Type 2 images. This indicates that Our-v8 has better feature extraction capabilities for coal blocks than YOLOv8.

The detection results are shown in Figure 10. As can be seen, under the illumination of a halogen light source, the color of the sample varies from black to grayish from left to right. The type-a sample has a clear luster and a distinct black color because the coalification degree of high-quality coal is high and the organic carbon content is high. The color of the type-c sample is grayish white because the coalification degree of gangue is low, resulting in low organic carbon content. While under the illumination of a fluorescent light source, the color differences between coal and gangue are not obvious. So, the coal of type-b is mistakenly identified as type-a by YOLOv8, as shown in the red-circle-marked area in Figure 10b, which means that YOLOv8 has difficulty distinguishing coal of type-a and type-b because of their extremely high similarity.

In addition, from the yellow-circle-marked areas in Figure 10a,c, it can be seen that the positioning performance of the Our-v8 prediction box is better than that of YOLOv8. This means that Our-v8 can provide more accurate location information for coal and gangue. In conclusion, based on the visualization results of the feature maps and the detection results, it can be seen that Our-v8, by introducing the CBAM attention mechanism and EIOU loss function, has better recognition and positioning capabilities than YOLOv8.

For example, the confidence and recognition frame position information of coal and gangue in the second row of Figure 10 are shown in Table 2, where (x_t_, y_t_) is the top-left coordinate of the recognition frame, (x_b_, y_b_) is the bottom-right coordinate of the recognition frame, and Label is the type of the identified target. By using the two coordinate values, we can obtain the relative position of the entire coal or gangue relative to the image, which is beneficial to the subsequent coal quality grading.

### 3.3. Model Test

The test results on the test dataset are shown in Table 3, and the sample number is the number of coal-a, coal-b, and gangue, respectively, in the images of the test dataset (shown in Table 1). It can be seen that the accuracy P and recall R of Our-v8 for the coal and gangue of Type 1 are all above 99.1%, and the average precision AP is also above 99.4%.

The confusion matrix (Figure 11) shows that only a few samples are misclassified. The average precision AP of Our-v8 for gangue of Type 2 is 98.5%, while the average precision AP for coal of type-a and type-b is only 93.5% and 91.6%, respectively. At the same time, it can also be seen in Figure 11b that 20 samples of type-a are misclassified as type-b, and 30 samples of type-b are misclassified as type-a. This means that Our-v8 is prone to confusing type-a and type-b in fluorescent light environments. In conclusion, Our-v8 can achieve a better identification effect under the illumination of a halogen light source than in a fluorescent light source environment.

### 3.4. Comparison with Other Advanced Models

To apply the model to the coal sorting conveyer in practice, the model must meet the requirements of high accuracy, fast detection speed, and lightweight. So, Our-v8 was compared with previous YOLO series models on the test dataset, as shown in Table 4. On the Type 1 test dataset, Our-v8’s mAP reached 99.5%, which is 14.9%, 10.2%, 7.4%, 3.6%, and 2% higher than YOLOv3, YOLOv4, YOLOv5, YOLOv7, and YOLOv8, respectively. Moreover, Our-v8 has lower FLOPs and Param than YOLOv3, YOLOv4, and YOLOv7. And on the Type 2 test dataset, Our-v8’s mAP is 12.7%, 8.3%, 6.3%, 3.6%, and 1.5% higher than YOLOv3, YOLOv4, YOLOv5, YOLOv7, and YOLOv8, respectively. There is no significant change in FLOPs and Param for the model.

In general, Our-v8 has obvious advantages over YOLO series models in accuracy and parameters. In addition, Our-v8 has a size of only 22.1 MB, and it takes only 8 ms to detect each image, with a detection speed of 125 FPS. This suggests that it has broad application potential in coal quality detection and sorting.

### 3.5. Comparison with Existing Research

Many studies have previously developed methods for sorting coal based on images. This study investigated and summarized these studies and compared them with our proposed method, as shown in Table 5.

As shown in Table 5, Luan et al. [5] proposed a coal and gangue classification method based on CNN. Compared with this model, the P and R of Our-v8 increased by 3.9% and 3.7%, respectively. Yan et al. [6] proposed an improved YOLOv5 model to detect coal and gangue. However, they used multispectral data as input to the model instead of RGB images, which means the detection workflow was more complex. Moreover, the mAP and detection speed of Our-v8 were 1.2% and 94.7 FPS higher than their model, respectively. Zhang et al. [1] used YOLOv4 to detect coal and gangue; the mAP and detection speed of Our-v8 were 2% and 114.8 FPS higher than theirs, respectively. Wen et al. [15] proposed a YOLOv5-Swin to detect coal and gangue, with an mAP of 98.6% and a detection speed of 147 FPS. However, the detection accuracy is still slightly lower than Our-v8. In addition, it is worth noting that the current methods basically only detect and classify coal and gangue. The proposed Our-v8 in this study can not only detect coal and gangue but also detect coal blocks of different densities. This means that the method proposed in this study can perform more refined coal sorting work. Moreover, Our-v8 has a high detection frame rate, which can meet the real-time requirements of mobile terminal detection.

## 4. Conclusions

This study aims to develop a lightweight coal and gangue detection model. To achieve this, a dataset is created by collecting samples from two coals of different densities and gangue under halogen lamp and fluorescent lamp lighting environments, and an improved deep learning model, Our-v8, is proposed. The results show that Our-v8 can detect coal and gangue in a halogen lamp lighting environment accurately with an mAP of 99.5%, FLOPs of 29.7, Param of 12.8, and a model size of only 22.1 MB. Additionally, Our-v8 can provide accurate location information for coal and gangue, making it ideal for real-time coal sorting applications. Overall, the Our-v8 proposed in this study has higher detection accuracy and speed, and the model has lightweight characteristics, making it highly promising for real-time coal sorting work.

## Figures and Tables

**Figure 1 sensors-24-01246-f001:**
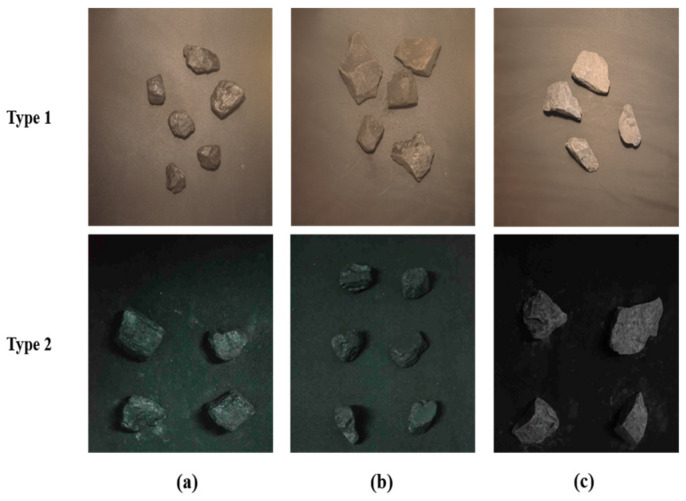
Dataset images: (**a**) coal with density < 1.4 g/cm^3^; (**b**) coal with 1.4 < density < 1.6; (**c**) gangue.

**Figure 2 sensors-24-01246-f002:**
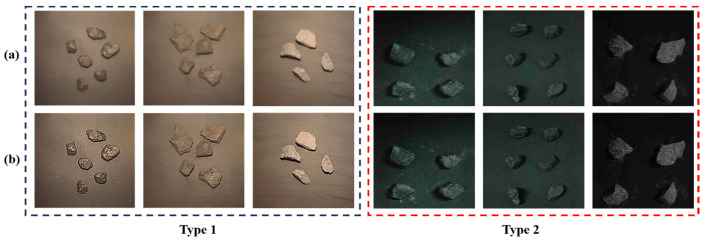
Comparison of images before and after Laplacian enhancement: (**a**) original image; (**b**) processed image.

**Figure 3 sensors-24-01246-f003:**
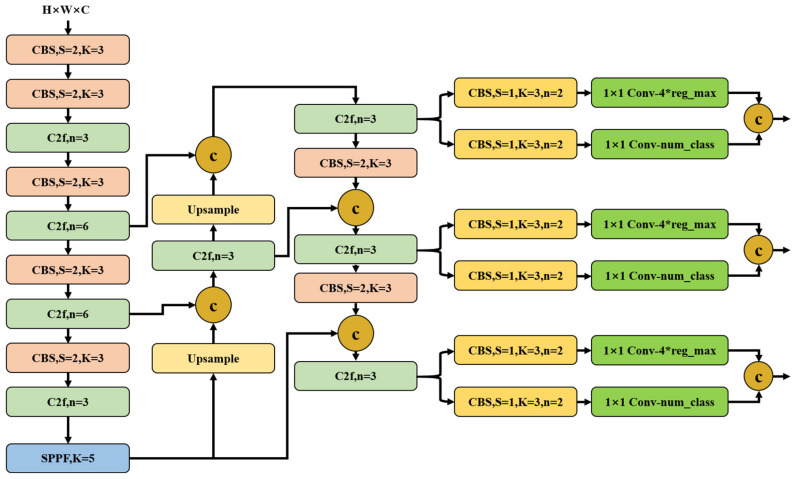
YOLOv8 Network Structure.

**Figure 4 sensors-24-01246-f004:**
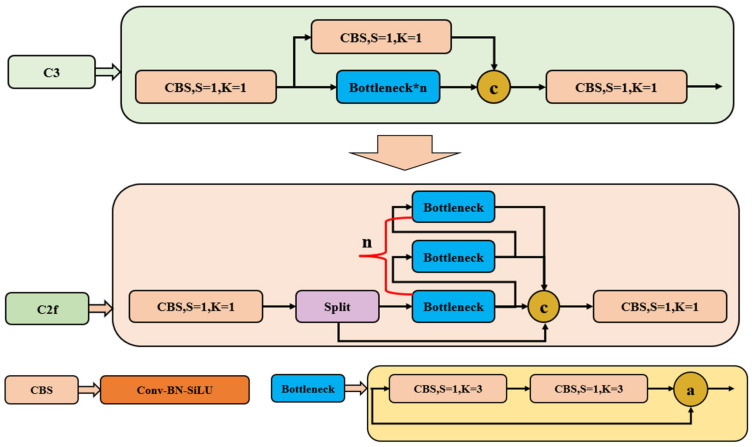
C2f feature extraction module.

**Figure 5 sensors-24-01246-f005:**
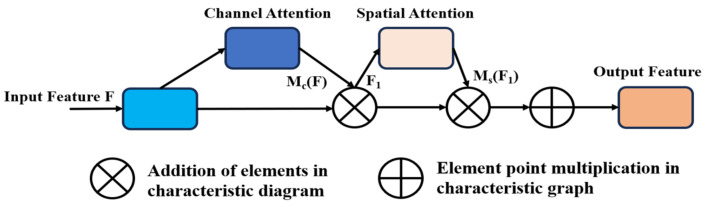
CBAM attention mechanism.

**Figure 6 sensors-24-01246-f006:**
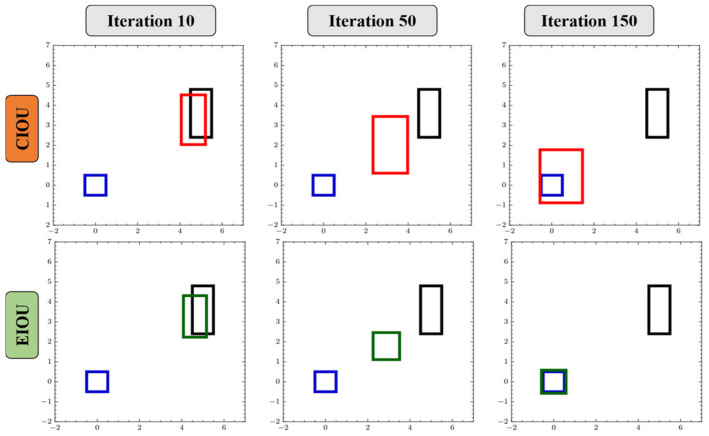
The prediction process of CIOU and EIOU.

**Figure 7 sensors-24-01246-f007:**
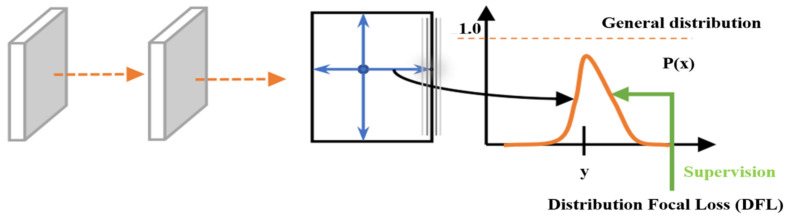
The distribution focal loss process.

**Figure 8 sensors-24-01246-f008:**
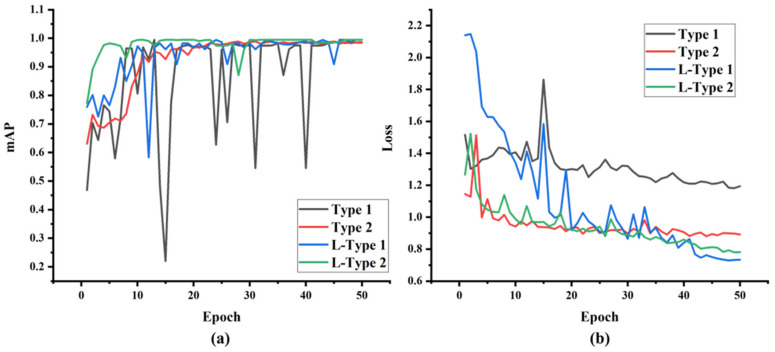
Model training process: (**a**) mAP curve; (**b**) loss curve.

**Figure 9 sensors-24-01246-f009:**
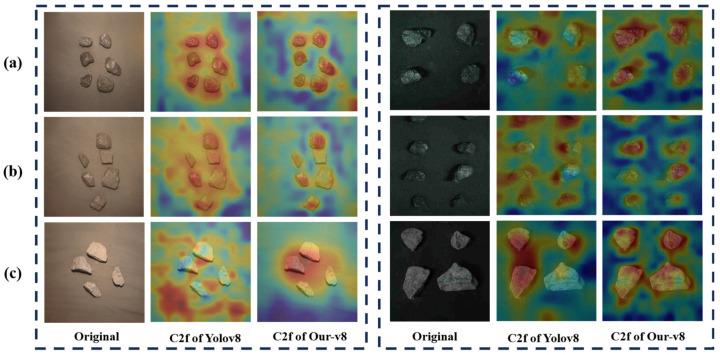
Feature maps of the C2f layer of YOLOv8 and Our-v8: (**a**) coal with density less than 1.4 g/cm^3^; (**b**) coal with density greater than 1.4 but less than 1.6 g/cm^3^; (**c**) gangue.

**Figure 10 sensors-24-01246-f010:**
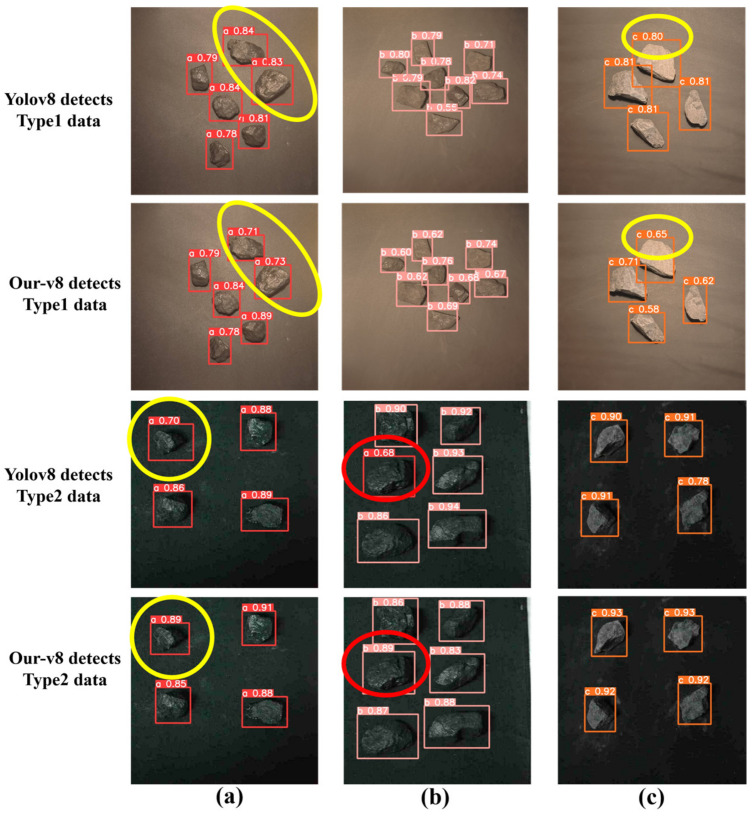
Detection results of YOLOv8 and Our-v8: (**a**) coal with density less than 1.4 g/cm^3^; (**b**) coal with density greater than 1.4 but less than 1.6 g/cm^3^; (**c**) gangue.

**Figure 11 sensors-24-01246-f011:**
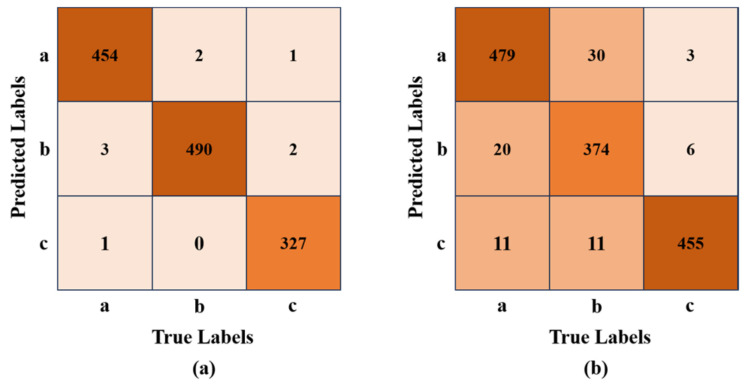
The confusion matrix of test results: (**a**) Type 1; (**b**) Type 2.

**Table 1 sensors-24-01246-t001:** Coal and gangue image dataset division.

		Training Set	Validation Set	Test Set
Type 1	total	891	127	256
	a	260	37	75
	b	373	53	107
	c	258	37	74
Type 2	total	962	138	277
	a	339	49	102
	b	296	43	83
	c	327	46	92

**Table 2 sensors-24-01246-t002:** Confidence and recognition frame position information of coal and gangue.

	Detection Result			
Serial Number	Label	Score	(x_t_, y_t_)/Pixel	(x_b_, y_b_)/Pixel
1	a	0.73	415, 202	548, 328
2	a	0.78	269, 454	340, 541
3	a	0.79	210, 200	258, 283
4	a	0.84	279, 298	370, 382
5	a	0.89	377, 398	464, 476
6	a	0.71	320, 172	445, 213
7	b	0.60	133, 179	217, 236
8	b	0.62	187, 262	294, 352
9	b	0.62	240, 118	309, 197
10	b	0.67	454, 254	566, 320
11	b	0.68	364, 266	437, 343
12	b	0.69	292, 363	394, 436
13	b	0.74	424, 151	514, 225
14	b	0.76	274, 209	361, 279
15	c	0.58	269, 454	340, 541
16	c	0.71	210, 200	258, 283
17	c	0.65	279, 298	370, 382
18	c	0.62	377, 398	464, 476

**Table 3 sensors-24-01246-t003:** Results of Our-v8 on the test dataset.

	Sample	Sample Number	P/%	R/%	AP/%
Type 1	a	458	99.3	99.1	99.4
	b	492	99.1	99.6	99.5
	c	330	99.4	99.2	99.5
Type 2	a	510	93.5	94	93.5
	b	415	93.9	90.3	91.6
	c	464	95.3	98.1	98.5

**Table 4 sensors-24-01246-t004:** Model comparison results.

Data Set	Model	mAP/%	FLOPs/G	Param/M	Volume/MB	FPS
Type 1	YOLOv3	84.6	154.9	61.3	117	39
	YOLOv4	89.3	30.5	67.4	245.5	42
	YOLOv5	92.1	22.4	4.3	14.4	138
	YOLOv7	95.9	104.3	37.1	284	96
	YOLOv8	97.5	28.5	11.1	21.4	128
	Our-v8	99.5	29.7	12.8	22.1	125
Type 2	YOLOv3	82.1	154.9	61.3	117	39
	YOLOv4	86.5	30.5	67.4	245.5	42
	YOLOv5	88.5	22.4	4.3	14.4	138
	YOLOv7	91.2	104.3	37.1	284	96
	YOLOv8	93.3	28.5	11.1	21.4	128
	Our-v8	94.8	29.7	12.8	22.1	125

**Table 5 sensors-24-01246-t005:** Performance comparison with other previous detection models.

Researcher	Model	P/%	R/%	mAP/%	FPS
Luan et al. (2023) [5]	CNN	95.4	94.6	×	×
Yan et al. (2022) [6]	Improved YOLOv5	×	×	98.3	30.3
Zhang et al. (2022) [1]	YOLOv4	×	×	97.5	10.2
Wen et al. (2023) [15]	YOLOv5-Swin	95.39	98.09	98.6	147
This Study	Our-v8	99.3	98.3	99.5	125

## Data Availability

Data are contained within the article.

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
