# Peer review of "Detection of Coal and Gangue Based on Improved YOLOv8"

_sensors, 2024, doi:10.3390/s24041246_

Round 1
Reviewer 1 Report
Comments and Suggestions for Authors
This authors proposed an innovative approach to address the challenges of lightweight and real-time coal sorting detection based on improved YOLOv8.They also focus on the detection of coal and gangue with different density and lighting variations,providing accurate location information.The manuscript is well-written and the approach is innovative. The study is meaningful for real-time coal sorting application.My recommendation is accept.The following is some minor sugessions.
1.Figure 9 should illustrats that the left and right images correspond to two types of lighting or types 1 and 2.
2.The caption of Figure 11 should illustrate (a) and (b).
3.Table2, The data in the fourth column should add a comma, and the same applies to the fifth column.Such as (xt, yt),the first line 415,202.
4.Pay attention on minor grammar mistakes.
Comments on the Quality of English LanguagePay attention on minor grammar mistakes.
Author Response
Thank you very much for your comments on this manuscript. Here is our reply to your comments:
Comment 1: Figure 9 should illustrate that the left and right images correspond to two types of lighting or types 1 and 2.
Response 1: Thanks for the comment. We have revised the sentence as follows, ‘the left part for dataset Type1 and the right part for Type2’, marked with red color in the manuscript.
Comment 2: The caption of Figure 11 should illustrate (a) and (b).
Response 2: Thanks for the comment. We have revised the sentence as follows. (a) Type1; (b) Type2, marked with red color in the manuscript.
Comment 3: Table2, The data in the fourth column should add a comma, and the same applies to the fifth column. Such as (xt, yt),the first line 415,202.
Response 3: Thanks for the comment. We have add comma in the fourth and fifth column of Tabel2, and marked red.
Comment 4: Pay attention on minor grammar mistakes.
Response 4: Thanks for the suggestion. We have checked the manuscript and correct the minor grammar mistakes, marked red.
Reviewer 2 Report
Comments and Suggestions for Authors
Thanks for your paper. I find it very thorough and methodical. It describes the proposed algorithm on a deep level which is always nice from a programming point of view. I have a few suggestions which could help the improve the paper further:
Some parts of the paper are very technical in the sense that abbreviations are used without first being introduced. This makes understanding what the authors did to optimized the model hard to follow without first searching for the explanations of these abbreviations in other literature. A more high level overview of the ideas behind the optimizations could help the reading to understand them faster, before the authors delve into the deeper details.
Lines 96-97: “To adapt to the model input, 96 the sizes of all images were cropped to 640×640 pixels.”
Is there a reason why you choose 640x640 pixels? Did you crop in the sense of cutting out a specific region of the original image or did you rescale the image?
Figure 1:
General question regarding the distribution of the coal and gangue chunks: in your images, all chunks are separated from each other, there are no touching objects and no small pebbles or dust. Is this a realistic scenario? In a mine I assume that the excavated raw material will be dumped on a conveyor belt without separating each chunk. Since coal is mined in bulk a realistic evaluation of the algorithm would be to use some images filled with coal and gangue as described with the final model. It is alright form a scientific point of view when you use semi-realistic images with separated chunks, but from an application viewpoint some examples with more realistic distributions would be nice.
Lines 132-134: “At the same time, DFL idea is used to extract more detailed information about the coals and gangues, the number of channels in the regression head is also changed to 4*reg_max to improve the regression accuracy.”
Could you state what DFL stands for before using it?
Lines 139-141: “As shown in Figure 4, the C2f module is designed based on the ideas of the C3 mod- ule and the ELAN, the idea of feature extraction from parallel streams in CSPNet, and combines the idea of residual structure, allowing YOLOv8 to obtain richer gradient flow information while ensuring lightweight.”
If the reader is not extremely familiar with the technical details of YOLO, C2f, C3 and ELAN will be confusing. Could you introduce the reader to these modules by first explaining what the abbreviations stand for and then shortly stating what the idea behind them is?
Lines 155-157: "In this study, the CBAM attention mechanism is embedded before the SPPF module, further enhancing the feature extraction ability of the backbone network, which can lead to improved accuracy in coal and gangues detection."
Same point as above, what is SPPF standing for and how does it work?
Lines 162-3: “The original box regression loss function of YOLOv8 is in the form of CIOU Loss + DFL.”
Same point as above, what is CIOU Loss standing for and how does it work?
Figure 8:
Regarding the Type 1 line - why are there several drastic drops in mAP even late in the training at epoch 40?
Comments on the Quality of English LanguageThere are some minor grammatical errors which can be solved easily.
Author Response
Thank you very much for your comments on this manuscript. Here is our reply to your comments:
Comment 1: Lines 96-97: “To adapt to the model input, 96 the sizes of all images were cropped to 640×640 pixels.” Is there a reason why you choose 640x640 pixels? Did you crop in the sense of cutting out a specific region of the original image or did you rescale the image?
Response 1: Thanks for the comment. Yolov8 model is designed for real-time inference and take an input size of 640x640. Using an input size different from the model's design size might impact accuracy and performance. So in this study, all the images were cropped as 640x640 to adapt to the yolov8. We cropped the original image, not rescale the image.
Comment 2: Figure 1:General question regarding the distribution of the coal and gangue chunks: in your images, all chunks are separated from each other, there are no touching objects and no small pebbles or dust. Is this a realistic scenario? In a mine I assume that the excavated raw material will be dumped on a conveyor belt without separating each chunk. Since coal is mined in bulk a realistic evaluation of the algorithm would be to use some images filled with coal and gangue as described with the final model. It is alright form a scientific point of view when you use semi-realistic images with separated chunks, but from an application viewpoint some examples with more realistic distributions would be nice.
Response 2: Thank you, this is a very good question. We also collected some images of coal gangue mixture overlapping in the laboratory. In the further study, we will take on-site photos of coal gangue in actual coal mine scenes, expand a large amount of dataset, and retrain our model for practical application.
Comment 3: Lines 132-134: “At the same time, DFL idea is used to extract more detailed information about the coals and gangues, the number of channels in the regression head is also changed to 4*reg_max to improve the regression accuracy.” Could you state what DFL stands for before using it?
Response 3: Thanks for the comment. DFL stands for Distribution Focal Loss. I have revised the manuscript and marked red.
Comment 4: Lines 139-141: “As shown in Figure 4, the C2f module is designed based on the ideas of the C3 module and the ELAN, the idea of feature extraction from parallel streams in CSPNet, and combines the idea of residual structure, allowing YOLOv8 to obtain richer gradient flow information while ensuring lightweight.”
If the reader is not extremely familiar with the technical details of YOLO, C2f, C3 and ELAN will be confusing. Could you introduce the reader to these modules by first explaining what the abbreviations stand for and then shortly stating what the idea behind them is?
Response 4: Thanks for the comment. CSPNet (Cross Stage Partial networks), YOLO(You Only Look Once), ELAN (Efficient Layer Aggregation Networks). C2f means CSPLayer_2Conv, C3 signifies a CSP module with three concatenation points and the ELAN module aims to strike a balance between feature richness and computational cost. I have revised the manuscript and marked red.
Comment 5:Lines 155-157: "In this study, the CBAM attention mechanism is embedded before the SPPF module, further enhancing the feature extraction ability of the backbone network, which can lead to improved accuracy in coal and gangues detection."
Same point as above, what is SPPF standing for and how does it work?
Response 5: Thanks for the comment. SPPF stands for Spatial Pyramid Pooling-Fast, SPPF takes the output feature maps from earlier layers in the YOLOv8 network as input and apply max pooling operations at different scales or "levels" of the spatial pyramid. I have revised the manuscript and marked red.
Comment 6: Lines 162-3: “The original box regression loss function of YOLOv8 is in the form of CIOU Loss + DFL.”
Same point as above, what is CIOU Loss standing for and how does it work?
Response 6: Thanks for the comment. CIOU Loss stands for Complete Intersection over Union Loss, aims to provide a more accurate and comprehensive measure of the similarity between the predicted bounding box and the ground truth box. By optimizing CIOU Loss during training, object detection models can achieve improved performance in terms of localization accuracy and stability. I have revised the manuscript and marked red.
Comment7:Figure 8:Regarding the Type 1 line - why are there several drastic drops in mAP even late in the training at epoch 40?
Response 7: Thanks for the comment. The several drastic drops in mAP of the Type 1 line maybe related to the learning rate, data distribution etc. After epoch 42, mAP is stable, so the training result is fine. In further study, we will add large amount of images in realistic scenario to our dataset, and try different training parameters to train our model.